# The effect of the universal test and treat strategy on the kidney function in adults living with HIV in Zambia: A six-month multicenter cohort study

Lukundo Siame[1]*, Matenge Mutalange[1], Chitalu Chanda[2], Morgan Sakala[3], Chilala Cheelo[1], Kingsley Kamvuma[1], Geofrey Mupeta[1], Martin Chakulya[1], Memory Ngosa[1], Michelo Haluuma Miyoba[1], Situmbeko Liweleya[1], Sepiso K. Masenga[1], Benson M. Hamooya[1]

**1** Mulungushi University School of Medicine and Health Sciences, Livingstone, Zambia, **2** Ministry of Health, Lusaka, Zambia, **3** Ministry of Health, Choma, Zambia

* lukundosiame23@gmail.com

## Abstract

### Background

Kidney disease is prevalent among people living with HIV (PLHIV), especially in Sub-Saharan Africa (SSA), due to complications of HIV infection, co-morbidities, and antiretroviral therapy (ART). Despite SSA shouldering a disproportionate burden of HIV, there is limited data on the effect of clinical and demographic factors on the kidney with the introduction of the Test and Treat policy. This study aimed to determine the incidence and factors associated with kidney impairment among PLHIV on ART in the Southern Province of Zambia.

### Methods

We conducted a retrospective cohort study among 1216 adult individuals living with HIV who initiated ART between January 1, 2014, and July 31, 2016 [before test-and-treat cohort (BTT), n = 814] and August 1, 2016, and October 1, 2020 [after test-and-treat cohort (ATT), n = 402] without kidney function impairment at baseline, followed for 6 months in 12 districts of the Southern Province. The primary outcome was kidney function impairment, defined by an estimated glomerular filtration rate (eGFR) of < 60 ml/min/1.73m² estimated using the Modification of Diet in Renal Disease (MDRD) equation. We used multivariable logistic regression (xtlogit model) to identify factors associated with kidney function impairment. Statistical significance was set at p < 0.05.

### Results

The median age was 36.4 years (interquartile range (IQR): 29.9, 43.3), and the majority of participants were women (57.2%, n = 695). Tenofovir Disoproxil Fumarate

**Data availability statement:** All relevant data are within the manuscript and its Supporting Information files.

**Funding:** The author(s) received no specific funding for this work.

**Competing interests:** The authors have declared that no competing interests exist.

(TDF) and XTC exposure was noted among 1,173/1216 (96.5%) enrolled participants and 92.9% (26/28)of those with renal impairment. The overall cumulative incidence of kidney impairment was 2.3% (n = 28/1216: 95% confidence interval (CI) 3%, 5%), and it was higher BTT compared to the ATT (2.8% vs. 1.2%). Every unit increase in age was associated with an increased odds of having kidney function impairment (adjusted odds ratio (AOR):1.05, 95% CI: 1.01–1.09, p = 0.008).. Participants from urban facilities also had a higher risk (AOR: 5.14, 95% CI: 1.95–13.55, p < 0.001). In contrast, being enrolled after the implementation of the "test-and-treat" policy was associated with lower odds of having kidney function impairment (AOR: 0.45, 95% CI: 0.12–0.97, p = 0.042).

## Conclusions

This study found a 2.3% incidence of kidney function impairment among PLHIV within 6 months of initiating ART. An increase in age and receiving care at an urban facility were positively associated with kidney function impairment, whereas ART enrollment following the implementation of the "test-and-treat" policy was negatively associated. This study highlights the benefits of early ART initiation on kidney function, reinforcing the need to maintain the universal test-and-treat policy.

## Introduction

Kidney disease is one of the most common non-communicable diseases among people living with HIV (PLHIV) globally, occurring in 3.5% to 48.5% depending on the geographical region, and sub-Saharan Africa bears a particularly high burden [1,2]. Due to the availability of effective antiretroviral therapy (ART), the number of people living with HIV is expected to rise, along with an increase in non-communicable diseases like kidney disease [2].

In Zambia, the burden of chronic kidney disease (CKD) among PLHIV ranges from 5% and 28% [3–5]. CKD in this population is linked to poor outcomes such as end-stage renal disease (ESRD) and mortality [3]. Zambia faces significant obstacles in managing CKD, with only one nephrologist per 2.6 million people and less than 10% of adults with ESRD having access to dialysis services [3]. Furthermore, the country lacks a kidney transplant program, forcing few patients who can manage to seek treatment abroad at huge costs [3]. These limitations severely hamper the ability to provide adequate care, contributing to high mortality rates [6].

Despite the high burden of kidney disease in our setting with endemic HIV, limited resources are allocated for routine and follow-up measurements for kidney function tests. Consequently, the true burden of kidney disease and its associated factors remain largely unknown. This gap highlights a missed opportunity to design targeted interventions and policies for addressing kidney impairment among PLHIV in resource-limited settings. Hence, it is crucial to quantify the burden of kidney disease and identify the associated factors using available routine data. Therefore, this

study aimed to determine the incidence of kidney impairment, the effect of the test-and-treat strategy, and the associated factors among adults living with HIV in the Southern Province of Zambia.

## Methods

### Study design and setting

This was a retrospective cohort study among PLHIV aged ≥15 years with a follow-up period of 6 months. It utilized data from a study focused on the clinical outcomes among individuals enrolled in HIV care and treatment before and after the Test-and-treat program [7]. The study was conducted across 12 districts in Zambia's Southern Province, including Chikankata, Choma, Kalomo, Kazungula, Livingstone, Mazabuka, Monze, Namwala, Pemba, Siavonga, Sinazongwe, and Zimba [7]. The data for this primary study was assessed from November 10, 2024, to December 19, 2024.

### Eligibility criteria

The primary study abstracted data from medical health records (SmartCare and paper-based health records) for individuals who initiated ART at a particular health facility and excluded anyone with missing information on date of birth or age. In this study, eligible participants were individuals with baseline serum creatinine results and without kidney impairment. We further excluded participants without creatinine results at 6 months, Fig 1.

### Variables in the study

The outcome variable was kidney impairment defined as having an estimated glomerular filtration rate (eGFR) of < 60 ml/min/1.73m$^2$ [8]. GFR was calculated using the modified four-variable MDRD formula [9]. The formula is as follows:

$$GFR = 175 \times (standardized\,SCr)\hat{}-1.154 \times (age)\hat{}-0.203 \times 1.212\,(if\,Black) \times 0.742(if\,female).$$

The independent variables included age, sex, facility, cohort, marital status, blood pressure (systolic and diastolic), viral load, CD4 count, WHO staging, ART regimen, height, and weight.

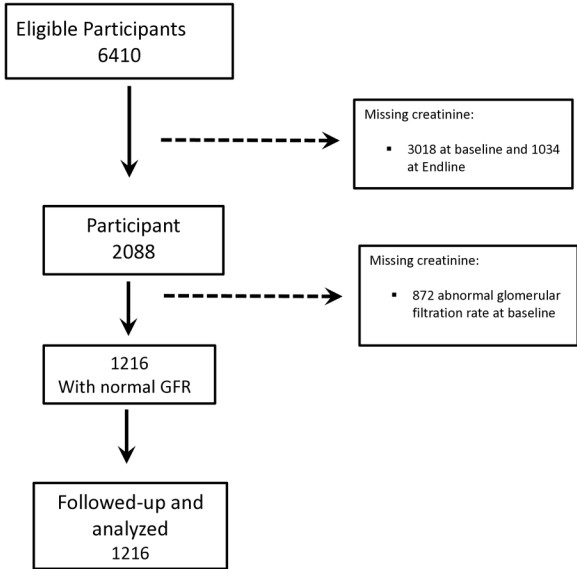

**Fig 1. Selection of participants.**

## Data collection

The primary study was conducted across 45 health facilities in 12 districts of Southern Province. Sociodemographic (age, sex, marital status), clinical (ART regimen, diastolic and systolic blood pressure, height, and weight), and laboratory (serum creatinine, CD4 count, and HIV viral load) data were abstracted from both paper-based and electronic HIV medical records (Smart Care) onto the Electronic Data Capture (REDCap) [7].

## Data analysis

Data were exported to Microsoft Excel for cleaning and thereafter analyzed in Stata version 15. Data were summarized using the frequencies and proportions for categorical variables and median (interquartile range) for continuous variables after testing for normality using the Shapiro-Wilk test and Q-Q plots. The Chi-square test was employed to determine statistical differences between two categorical variables, while the Wilcoxon rank-sum test was used to compare two independent medians. Multivariable logistic regression (xtlogit model) to account for the correlation structure of the data was used in this cohort study to determine factors associated with kidney impairment among the study participants. The characteristics in the final model were chosen based on previous literature [3–5]. To ascertain statistical significance, a p-value of $< 0.05$ was used.

## Ethical considerations

Ethical approval for the data used in this study was obtained from the Mulungushi University School of Medicine and Health Sciences (SOHMS) Research Ethics Committee (ethics
Reference number SMHS-MU2-2024-15) on 21st May 2024. Due to the use of secondary data, the need for informed participant consent was waived.

## Results

### Basic characteristics of the study participants

The median age of participants was 36.4 years (interquartile range [IQR]: 29.9–43.3). Individuals in the after test-and-treat (ATT) cohort were slightly older compared to those in the before test-and-treat (BTT) cohort (37.2 vs. 36.0 years, $p < 0.048$). The majority of participants in this study were women, accounting for 57.2% (n = 695). Most participants in both the BTT (n = 762, 93.6%) and ATT (n = 358, 89.1%) cohorts were from urban areas. There were more married individuals compared to unmarried individuals in both cohorts: 63.3% (n = 515) in BTT and 59.5% (n = 239) in ATT. Participants in the ATT cohort had slightly higher systolic and diastolic blood pressure compared to those in the BTT cohort (117 mmHg vs. 115 mmHg, $p < 0.015$; and 75 mmHg vs. 73 mmHg, $p < 0.004$, respectively). The TDF/XTC regimen was the major backbone combination among NRTIs, while non-nucleotide reverse transcriptase inhibitors (NNRTIs) were the most used companion drugs across all clients. In the BTT cohort, 99.8% of participants were on NNRTIs, compared to 95.8% in the ATT cohort. Similarly, 96.1% of participants in the BTT cohort and 97.5% in the ATT cohort were on TDF/XTC. The median duration on ART was significantly longer in the BTT cohort compared to the ATT cohort (18 months vs. 5.5 months, $p < 0.001$). At baseline, the majority of patients were classified as WHO Stage I HIV, with a higher proportion of Stage I patients in the BTT cohort (n = 577, 74.1%) compared to the ATT cohort (n = 220, 57.4%). Creatinine levels at baseline were significantly higher in the ATT cohort (66 μmol/L) compared to the BTT cohort (60.5 μmol/L, $p < 0.001$). The estimated Glomerular Filtration Rate (eGFR) at baseline were significantly higher in the ATT cohort compared to the BTT cohort (132.5 vs 128, $P < 0.001$) while eGFR at 6 months was higher in the BTT compared to the ATT (123 vs. 111). The median CD4 count was similar between the groups (274 cells/μL in BTT vs. 279 cells/μL in ATT, $p = 0.705$). A slightly higher proportion of participants in the BTT cohort were virally suppressed compared to the ATT cohort (89.1% vs. 82.1%, respectively; see Table 1).

**Table 1. Demographic and clinical characteristics sorted by treatment policy status.**

| Variable | Before test-and-treat n = 814 | After test-and-treat n = 402 | P < value |
|---|---|---|---|
| Age* | 36.0 (29.7, 42.8) | 37.2 (31, 44) | **0.048** |
| Sex ** | | | 0.602 |
| Male | 353 (43.4) | 168 (41.8) | |
| Female | 461 (56.6) | 234 (58.2) | |
| Marital status ** | | | 0.197 |
| Married | 515 (63.3) | 239 (59.5) | |
| Unmarried | 299 (36.7) | 163 (40.6) | |
| Facility | | | **0.006** |
| Rural | 52 (6.4) | 44 (11.0) | |
| Urban | 762 (93.6) | 358 (89.1) | |
| Systolic blood pressure, mmHg* | 115 (104, 127) | 117 (106, 131) | **0.015** |
| Diastolic blood pressure, mmHg* | 73 (65, 81) | 75 (68, 85) | **0.004** |
| Weight, kg * | 58 (52, 66) | 58 (52, 66) | 0.189 |
| Height, cm * | 164 (159, 170) | 165 (159, 171) | 0.820 |
| Body mass index, kg/m2** | | | 0.801 |
| Underweight | 142 (18.3) | 61 (16.1) | |
| Normal | 475 (61.2) | 237 (62.4) | |
| Overweight | 115 (14.8) | 58 (15.3) | |
| Obesity | 44 (5.7) | 24 (6.3) | |
| ART ** | | | **< 0.001** |
| NNRTI (EFV & NVP) | 811 (99.8) | 385 (95.8) | |
| PI (LPV/r & ATZ/r) | 0 (0.0) | 14 (3.5) | |
| INSTI (DTG) | 2 (0.3) | 3 (0.8) | |
| NRTI | | | 0.531 |
| ABC/3TC | 27 (3.3) | 8 (2.0) | |
| AZT/3TC | 4 (0.5) | 2 (0.5) | |
| D4T/3TC | 1 (0.1) | 0 (0.0) | |
| TDF/ XTC | 781 (96.1) | 392 (97.5) | |
| WHO clinical staging of HIV ** | | | **< 0.001** |
| 1 | 577 (74.1) | 220 (57.4) | |
| 2 | 117 (15.0) | 117 (30.6) | |
| 3 | 75 (9.7) | 41 (10.7) | |
| 4 | 10 (1.3) | 5 (1.3) | |
| Median duration on ART, months | 18 (13, 42) | 5.5 (0, 15) | **< 0.001** |
| Viral load, copies/ml ** | | | |
| CD4 count, cells/µL* | 274 (154, 443) | 279 (147, 455) | 0.705 |
| Virally suppressed ** | | | 0.082 |
| Yes | 236 (89.1) | 78 (82.1) | |
| No | 29 (10.9) | 17 (17.9) | |
| Creatinine at baseline, µmol/L* | 60.5 (50, 76.8) | 66 (56, 80) | **< 0.001** |
| Creatinine at 6 months, µmol/L* | 64 (54, 77) | 65 (54, 77.5) | 0.459 |
| eGFR at baseline, mL/min/1.73 m²* | 128 (101, 159) | 132.5 (105,167) | **< 0.001** |
| eGFR at 6 months, mL/min/1.73 m²* | 123 (97,152) | 111 (89, 137) | **< 0.001** |

**Abbreviation:**

*data presented as median (interquartile range),

**data presented as frequency (%), NNRTI non-nucleoside/nucleotide reverse transcriptase inhibitor (EFV = efavirenz and NVP = Nevirapine), PI protease inhibitor (LPV/r = lopinavir/ritonavir and ATV/r = atazanavir/ritonavir), INSTI integrase strand transfer inhibitor (DTG = dolutegravir), ABC/3TC: Abacavir/ Lamivudine AZT/3TC: Zidovudine/ Lamivudine, D4T/3TC (Stavudine/ Lamivudine), TDF/XTC (Tenofovir Disoproxil Fumarate/ Emtricitabine or Lamivudine (XTC refers to either Emtricitabine or Lamivudine depending on the formulation), eGFR (estimated Glomerular Filtration Rate)

## Kidney impairment in relation with other study variables and prevalence

Table 2 illustrates the relationship between kidney impairment with other study variables. The cumulative incidence of kidney impairment was 2.3% (28/1216; 95% CI: 1.5%–3.0%), with a higher proportion of cases observed in urban areas compared to rural areas (75.0% vs. 25.0%). Individuals with kidney impairment were older in comparison with those without kidney impairment, 38.6 vs. 36.3 years with a borderline p-value of 0.061. A higher proportion of individuals who initiated ART before the implementation of the test- and-treat policy had kidney impairment, 82.1% vs. 17.9%.

## Regression analysis of factors associated with kidney impairment

Table 3 shows the results of univariable and multivariable regression analyses of factors associated with kidney impairment. At univariable analysis, participants who were older had 1.04 times higher odds of having kidney impairment than those without (Odds ratio (OR): 1.04, 95% CI: 1.01–1.08, p = 0.011). Participants from urban facilities had 4.12 higher odds of having kidney impairment compared to those from rural facilities (OR: 4.12, 95% CI: 0.70–9.95, p = 0.002).

At multivariable analysis, for every unit increase in age, the older participants had an increased chance of 5% of having kidney impairment compared to those without (adjusted odds ratio (AOR): 1.05, 95% CI: 1.01–1.09, p = 0.008). Individuals who initiated ART after the implementation of the "Test and Treat" policy had a 55% reduced chance of having kidney impairment in comparison to those who initiated before the implementation of the policy (AOR: 0.45, 95% CI: 0.12–0.97, p = 0.042). Participants from urban heath facilities were 5.14 times more likely to have kidney impairment compared to those from rural health facilities (AOR: 5.96, 95% CI: 1.95–13.55, p < 0.001).

## Discussion

The study found the cumulative incidence of kidney impairment of 2.3% among PLHIV who were initiated on ART at 6 months of care. This incidence is similar to a study done in south Africa (2016), of which 2.3% of the participants had kidney impairment [10]. The incidence observed in our study raises significant concerns for individuals living with HIV, as kidney impairment in this population is strongly associated with a higher risk of end-stage renal disease (ESRD), and increased overall mortality [11]. Furthermore, it poses therapeutic challenges, as many antiretroviral drugs are metabolized or excreted by the kidneys, heightening the risk of drug toxicity and kidney injury [12,13]. Even though this cohort has treatment options like dialysis and transplants, they often face difficulty accessing them in resource-limited countries like ours, a challenge largely driven by issues such as limited clinic space, overcrowding, and shortages of clinical staff for screening and referrals, highlighting the need for data-driven interventions and policies for people with HIV[5,11].

In this current study, we observed a correlation between increasing age and kidney impairment among individuals living with HIV. This aligns with previous studies conducted by Penner et al. (2023) and Doshi et al. (2019) [14,15]. The observed association may be explained by the increased prevalence of weakened immune systems, comorbidities, and prolonged exposure to HIV with advancing age [16,17]. These factors can contribute to kidney damage as the virus directly infects kidney cells, leading to inflammation, fibrosis, scarring, hardening of kidney filtering units, and abnormal kidney proliferation of kidney cells, thus leading to quick progressive kidney impairment function, especially among older patients [17–19]. In addition, most of the participants were on Tenofovir Disoproxil Fumarate (TDF), a drug that damages the kidneys primarily by accumulating in proximal tubule cells and causing mitochondrial toxicity [20,21]. This can result to Fanconi syndrome and long-term kidney fibrosis and ultimately reduced Glomerular Filtration Rate (GFR) in some patients, mostly within the first years of exposure to the drug [21,22]. This finding suggests the need for routine screening of kidney functionality, especially among adults living with HIV, so that proper management is given to reduce poor outcomes.

Participants enrolled after the introduction of the universal test- and-treat policy had significantly reduced risk of having kidney impairment at 6 months. Despite the limited studies looking at the association between test- and-treat policy and kidney

**Table 2. Relationship between baseline demographic and clinical characteristic and kidney impairment.**

| Variable | kidney impairment | | |
|---|---|---|---|
| | Yes = 28 (2.3%) | No = 1,188 (97.7%) | P-value |
| Age* | 38.6 (33.4, 53.2) | 36.3 (29.9,43.3) | 0.061 |
| Sex ** | | | 0.698 |
| Male | 13(46.4) | 508 (42.8) | |
| Female | 15 (53.6) | 680 (57.2) | |
| Marital status ** | | | 0.887 |
| Married | 17 (60.7) | 737 (62.0) | |
| Unmarried | 11 (39.3) | 451 (38.0) | |
| Cohort | | | 0.084 |
| After Test and treat | 5 (17.9) | 397 (33.4) | |
| Before test and Treat | 23 (82.1) | 791 (66.6) | |
| Residence | | | **< 0.001** |
| Rural | 7 (25.0) | 89 (7.5) | |
| Urban | 21 (75.0) | 1099 (92.5) | |
| Systolic blood pressure, mmHg* | 114 (110.5, 124.5) | 116 (104, 129) | 0.879 |
| Diastolic blood pressure, mmHg* | 76.5 (63.5, 80.5) | 74 (66, 83) | 0.764 |
| Weight, kg * | 58.5 (49.5, 69.5) | 58 (52, 66) | 0.706 |
| Height, cm * | 166 (155,171) | 164 (159, 170) | 0.924 |
| Body mass index, kg/m2** | | | 0.106 |
| Underweight | 5 (18.5) | 198 (17.5) | |
| Normal | 17 (63.0) | 695 (61.6) | |
| Overweight | 1 (3.7) | 172 (15.2) | |
| Obesity | 4 (14.8) | 64 (5.7) | |
| ART ** | | | 0.796 |
| NNRTI (EFV & NVP) | 28 (100) | 1168 (98.4) | |
| PI (LPV/r & ATZ/r) | 0 (0.0) | 14 (1.2) | |
| INSTI (DTG) | 0 (0.0) | 5 (0.4) | |
| NRTI** | | | 0.132 |
| ABC/3TC | 1 (3.6) | 34 (2.9) | |
| AZT/3TC | 1 (3.5) | 5 (0.4) | |
| D4T/3TC | 0 (0.0) | 1 (0.1) | |
| TDF/ XTC | 26 (92.9) | 1147 (96.6) | |
| WHO clinical staging of HIV ** | | | 0.565 |
| 1 | 21 (80.8) | 776 (68.3) | |
| 2 | 3 (11.5) | 231 (20.3) | |
| 3 | 2 (7.7) | 114 (10.0) | |
| 4 | 0 (0.0) | 15 (1.3) | |

Abbreviation.

*Data presented as median (interquartile range),

**data presented as NNRTI NNRTI non-nucleoside/nucleotide reverse transcriptase inhibitor (EFV = efavirenz and NVP = Nevirapine), PI protease inhibitor (LPV/r = lopinavir/ritonavir and ATV/r = atazanavir/ritonavir), INSTI integrase strand transfer inhibitor (DTG = dolutegravir), ABC/3TC: Abacavir/ Lamivudine AZT/3TC: Zidovudine/ Lamivudine, D4T/3TC (Stavudine/ Lamivudine), TDF/XTC (Tenofovir Disoproxil Fumarate/ Emtricitabine or Lamivudine (XTC refers to either Emtricitabine or Lamivudine depending on the formulation)

**Table 3. Logistic regression analysis of baseline demographic and clinical characteristics associated with kidney impairment.**

| Variable | Univariable analysis | | Multivariable analysis | |
|---|---|---|---|---|
| | OR (95%CI) | P-value | OR (95%CI) | P-value |
| Age* | 1.04 (1.01, 1.08) | **0.011** | 1.05 (1.01, 1.09) | **0.008** |
| Sex ** | | | | |
| Male | REF | | REF | |
| Female | 0.86 (0.41, 0.83) | 0.699 | 1.02 (0.42, 2.42) | 0.968 |
| Marital status ** | | | | |
| Unmarried | REF | | | |
| Married | 0.95 (0.44 2.04) | 0.887 | 0.84 (0.36, 1.94) | 0.687 |
| Cohort | | | | |
| Before test and Treat | REF | | REF | |
| After Test and treat | 0.43 (0.16, 1.15) | 0.092 | 0.45 (0.12, 0.97) | **0.042** |
| Facility | | | | |
| Rural | REF | | REF | |
| Urban | 4.12 (0.70, 9.95) | **0.002** | 5.14 (1.95, 13.55) | **< 0.001** |
| Systolic blood pressure, mmHg* | 1.00 (0.99, 1.02) | 0.602 | 1.02 (0.98, 1.05) | 0.276 |
| Diastolic blood pressure, mmHg* | 0.99(0.96,1.02) | 0.51 | 0.97 (0.92, 1.01) | 0.140 |
| Body mass index, kg/m2** | | | | |
| Underweight | REF | | REF | |
| Normal | 0.97 (0.35, 2.66) | 0.951 | 0.84 (0.29, 2.40) | 0.744 |
| Overweight | 0.23 (0.03, 1.99) | 0.182 | 0.19 (0.02, 1.76) | 0.145 |
| Obesity | 2.48 (0.65,9.50) | 0.187 | 2.34 (0.51, 10.68) | 0.271 |
| WHO clinical staging of HIV ** | | | | |
| 1 | REF | | REF | |
| 2 | 0.48 (0.14, 1.62) | 0.238 | 0.61 (0.17, 2.14) | 0.439 |
| 3 | 0.65 (0.15, 2.80) | 0.562 | 0.55 (0.12, 2.57) | 0.455 |
| 4 | | | | |

Abbreviation.

*Data presented as median (interquartile range),

**data presented as NNRTI non-nucleoside/nucleotide reverse transcriptase inhibitor (EFV = efavirenz and NVP = Nevirapine), PI protease inhibitor (LPV/r = lopinavir/ritonavir and ATV/r = atazanavir/ritonavir), INSTI integrase strand transfer inhibitor (DTG = dolutegravir), NRTI nucleotide), NRTI nucleotide reverse transcriptase inhibitor.

impairment, in general test- and-treat policy has been shown to improve clinical outcomes in patients as evidenced by the study in Zambia (2023) [7]. Individuals enrolled before the test-and-treat policy often initiated ART with advanced HIV stages as seen in our study, at which point the virus had likely already caused kidney damage, including podocyte dysfunction, glomerular basement membrane thickening, and tubulointerstitial inflammation and fibrosis, leading to HIV-associated nephropathy, as well as an increased risk of Immune Reconstitution Inflammatory Syndrome (IRIS), which can also cause kidney injury [7,23]. Early initiation of antiretroviral therapy (ART) through test-and-treat can potentially mitigate these detrimental effects.

Participants from urban areas were five times more likely to experience kidney impairment when compared to those from rural areas. However, this finding may just reflect the disproportionately higher number of urban participants in the study. Additionally, in our setting, creatinine testing is more commonly conducted in urban treatment centers than in rural areas. while urbanization has been linked to lifestyle factors such as reduced physical activity and increased consumption of highly processed, energy-dense foods, driving the prevalence of hypertension, diabetes, and obesity, all of which are

key risk factors for kidney impairment [24,25]. Further studies may be needed to explore the underlying factors contributing to kidney impairment among people with HIV in urban settings.

The study has strengths and weaknesses. This was a retrospective study, thus important variables that might affect kidney function, such as alcohol, diabetes, smoking, and hepatitis, were not collected. Additionally, the exact point at which kidney injury occurred could not be determined. However, this study has advantages; it has a large sample size and adds to the growing literature on non-communicable diseases among adults living with HIV in a resource-limited setting.

## Conclusion

This study revealed a significant number of PLHIV who developed kidney impairment within a space of six months after ART initiation. Older age and receiving care at an urban facility were associated risk factors, while initiating ART after the implementation of the test-and-treat was a protective factor. These findings highlights the need for balanced allocation of limited resources directed to regular kidney function screening in PLHIV and the crucial role of early ART initiation in improving clinical outcomes.

## Supporting information

**S1 Checklist. Strobe check list.**
(DOCX)

**S2 Dataset. Mininal dataset.**
(XLSX)

## Acknowledgments

We would like to extend our gratitude to the HAND group chaired by Professor Masenga and Dr Hamooya for their support.

## Author contributions

**Conceptualization:** Lukundo Siame, Benson M. Hamooya.

**Data curation:** Benson M. Hamooya.

**Formal analysis:** Lukundo Siame, Sepiso K. Masenga, Benson M. Hamooya.

**Funding acquisition:** Benson M. Hamooya.

**Investigation:** Benson M. Hamooya.

**Methodology:** Lukundo Siame, Benson M. Hamooya.

**Project administration:** Benson M. Hamooya.

**Resources:** Benson M. Hamooya.

**Software:** Benson M. Hamooya.

**Supervision:** Benson M. Hamooya.

**Validation:** Morgan Sakala, Benson M. Hamooya.

**Visualization:** Lukundo Siame, Matenge Mutalange, Chitalu Chanda, Morgan Sakala, Chilala Cheelo, Kingsley Kamvuma, Martin Chakulya, Geofrey Mupeta, Memory Ngosa, Michelo Haluuma Miyoba, Situmbeko Liweleya, Sepiso K. Masenga, Benson M. Hamooya.

**Writing – original draft:** Lukundo Siame, Benson M. Hamooya.

**Writing – review & editing:** Lukundo Siame, Matenge Mutalange, Chitalu Chanda, Morgan Sakala, Chilala Cheelo, Kingsley Kamvuma, Situmbeko Liweleya, Sepiso K. Masenga, Benson M. Hamooya.

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
