## [Decision Letter · Decision Letter 0]

18 Mar 2025

PONE-D-25-05682Universal Test and Treat strategy effect on Kidney Function in Adults living with HIV in Zambia: A Six-Month Multicenter Retrospective Cohort StudyPLOS ONE

Dear Dr. Siame,

Thank you for submitting your manuscript to PLOS ONE. After careful consideration, we feel that it has merit but does not fully meet PLOS ONE’s publication criteria as it currently stands. Therefore, we invite you to submit a revised version of the manuscript that addresses the points raised during the review process.

The reviewers have made several suggestions for improvement, including pointing out contradicting data in different sections of the manuscript. Kindly review and resolve the issues raised.

We look forward to receiving your revised manuscript.

Kind regards,

Chika Kingsley Onwuamah, Ph.D.

Academic Editor

PLOS ONE

Journal Requirements:

2. Please include a copy of Table 1, which you refer to in your text on page 5.

3. Please remove all personal information, ensure that the data shared are in accordance with participant consent, and re-upload a fully anonymized data set.

Additional guidance on preparing raw data for publication can be found in our Data Policy (https://journals.plos.org/plosone/s/data-availability#loc-human-research-participant-data-and-other-sensitive-data ) and in the following article: http://www.bmj.com/content/340/bmj.c181.long.

Reviewers' comments:

Reviewer's Responses to Questions

**Comments to the Author**

1. Is the manuscript technically sound, and do the data support the conclusions?

Reviewer #1: Partly

Reviewer #2: Yes

Reviewer #3: Yes

2. Has the statistical analysis been performed appropriately and rigorously? 

Reviewer #1: Yes

Reviewer #2: Yes

Reviewer #3: Yes

3. Have the authors made all data underlying the findings in their manuscript fully available?

Reviewer #1: Yes

Reviewer #2: Yes

Reviewer #3: Yes

4. Is the manuscript presented in an intelligible fashion and written in standard English?

Reviewer #1: Yes

Reviewer #2: Yes

Reviewer #3: Yes

5. Review Comments to the Author

Reviewer #1: The authors have reported their research findings in the manuscript titled "Universal Test and Treat strategy effect on Kidney Function in Adults living with HIV in Zambia: A Six-Month Multicenter Retrospective Cohort Study". I have the following comments/observations:

1. I suggest the authors should consider rephrasing the title to read "The effect of the universal test and treat strategy on the kidney function in adults living with HIV in Zambia: A six-month multi center cohort study".

2. The eGFR calculation formulae used as reported were CKD-EPI in the abstract, and MDRD in the body of the manuscript. The authors should clarify the formula that was actually used.

3. In the results section, basic characteristics of study participants (page 5, line 11), the authors mentioned that "a high proportion of participants were taking NNRTI and ABC/3TC regimens. This is similar to what is reported in table 1; however, in table 2, the authors reported 1,173 participants on TDF/XTC, while only 35 participants were on ABC/3TC. The authors need to resolve this discordant results.

4. The explanation by the authors to the fact that in the ATT era, participants commenced ART earlier, and did not have renal complications associated with HIV makes scientific sense. However, TDF-based regimens have been reported extensively in literature to be associated with renal impairment in older patients, patients with co-morbidities etc. So it is critical for the authors to clarify whether these cohorts of patients were predominantly on ABC/3TC or TDF/XTC.

Reviewer #2: Reviewer’s comments on the manuscript titled “ Universal Test and Treat strategy effect on Kidney Function in Adults living with HIV in Zambia: A Six-Month Multicenter Retrospective Cohort Study.”

1. Originality of value:

The study is original and will contribute to scientific knowledge and enrich the repository of knowledge on burden of kidney disease among patient living with HIV, its risk factors and how it can be prevented or detected early, particularly in the sub-Saharan African region. The abstract is concisely written while the introduction was appropriately described, and study objective was clearly defined.

2. Suitability and Soundness:

The methodology is appropriate and suitable for the study design.

3. Clarity of presentation:

The manuscript presentation and lay out is clear and easy to follow.

4. Areas requiring corrections:

None

Reviewer #3: This article about HIV and kidney disease is timely and pertinent because it highlights the potential for preventing kidney disease. The drawback is the use of eGFR as the sole indicator of kidney impairment. An important hallmark of kidney damage is proteinuria whci was not utilised in this patient. One understands the operational issues with regards to HIV treatment funding and implementation especially with laboratory tests. But if serum creatinine could be done, microalbuminuria/proteinuria could be done.

Older patients would have decline in kidney impairment as per eGFR estimation. Could could the authors provide summary on

of eGFR in the two groups in addition to using it as a cut-off point. Could the trend analysis be done since this since a cohort?

The explanation for kidney impairment among urban dwellers is far fetched, especially when this is not based on the data from the study. There is no evidence that these patients had hypertension or diabetes. It may just reflect the disproportionately large number of urban dwellers in the study.

6. PLOS authors have the option to publish the peer review history of their article (what does this mean? ). If published, this will include your full peer review and any attached files.

**Do you want your identity to be public for this peer review?** For information about this choice, including consent withdrawal, please see our Privacy Policy .

Reviewer #1: **Yes: ** Professor Oche O. Agbaji

Reviewer #2: **Yes: ** Yemi R. Raji

Reviewer #3: No

---

## [Author Response · Author response to Decision Letter 0]

24 Mar 2025

Review Comments to the reviewer.

Reviewer #1: The authors have reported their research findings in the manuscript titled "Universal Test and Treat strategy effect on Kidney Function in Adults living with HIV in Zambia: A Six-Month Multicenter Retrospective Cohort Study". I have the following comments/observations:

1. I suggest the authors should consider rephrasing the title to read "The effect of the universal test and treat strategy on the kidney function in adults living with HIV in Zambia: A six-month multi center cohort study".

Response: thank you for the suggestion. We have changed the title.

2. The eGFR calculation formulae used as reported were CKD-EPI in the abstract, and MDRD in the body of the manuscript. The authors should clarify the formula that was actually used.

Response: thank you for the suggestion. We have changed the title.

3. In the results section, basic characteristics of study participants (page 5, line 11), the authors mentioned that "a high proportion of participants were taking NNRTI and ABC/3TC regimens. This is similar to what is reported in table 1; however, in table 2, the authors reported 1,173 participants on TDF/XTC, while only 35 participants were on ABC/3TC. The authors need to resolve these discordant results.

Response: thank you for your observations. We have corrected this discordant results in the table.

4. The explanation by the authors to the fact that in the ATT era, participants commenced ART earlier, and did not have renal complications associated with HIV makes scientific sense. However, TDF-based regimens have been reported extensively in literature to be associated with renal impairment in older patients, patients with co-morbidities etc. So it is critical for the authors to clarify whether these cohorts of patients were predominantly on ABC/3TC or TDF/XTC.

Response: thank you. we have clarified in the results section and discussion section that most of participants in this study were on TDF/XTC and further discussed the results.

Reviewer #2: Reviewer’s comments on the manuscript titled “Universal Test and Treat strategy effect on Kidney Function in Adults living with HIV in Zambia: A Six-Month Multicenter Retrospective Cohort Study.”

1. Originality of value:

The study is original and will contribute to scientific knowledge and enrich the repository of knowledge on burden of kidney disease among patient living with HIV, its risk factors and how it can be prevented or detected early, particularly in the sub-Saharan African region. The abstract is concisely written while the introduction was appropriately described, and study objective was clearly defined.

2. Suitability and Soundness:

The methodology is appropriate and suitable for the study design.

3. Clarity of presentation:

The manuscript presentation and layout is clear and easy to follow.

4. Areas requiring corrections:

None

Response: thank you for appraisal of the manuscript.

Reviewer #3: This article about HIV and kidney disease is timely and pertinent because it highlights the potential for preventing kidney disease. The drawback is the use of eGFR as the sole indicator of kidney impairment. An important hallmark of kidney damage is proteinuria whci was not utilised in this patient. One understands the operational issues with regards to HIV treatment funding and implementation especially with laboratory tests. But if serum creatinine could be done, microalbuminuria/proteinuria could be done.

Older patients would have decline in kidney impairment as per eGFR estimation. Could could the authors provide summary on

of eGFR in the two groups in addition to using it as a cut-off point. Could the trend analysis be done since this since a cohort?

Response: thank you. We have now provided the summary of eGFR at baseline and at the endline of the study. Unfortunately, we are unable to do the trend analysis due to the nature in which the study was conducted. We only collected data at two points at baseline of the study and at 6 months. In addition, our result in tables two and three point to the fact that older individuals were at risk of kidney impairment.

The explanation for kidney impairment among urban dwellers is far-fetched, especially when this is not based on the data from the study. There is no evidence that these patients had hypertension or diabetes. It may just reflect the disproportionately large number of urban dwellers in the study.

Response: thank you. We have revised the explanation of as observed, indeed this was might have been a reflection due to disproportionately large numbers of urban dwellers in this study.

---

## [Decision Letter · Decision Letter 1]

11 Apr 2025

The effect of the universal test and treat strategy on the kidney function in adults living with HIV in Zambia: A six-month multi center cohort study

PONE-D-25-05682R1

Dear Dr. Siame,

We’re pleased to inform you that your manuscript has been judged scientifically suitable for publication and will be formally accepted for publication once it meets all outstanding technical requirements.

Kind regards,

Chika Kingsley Onwuamah, Ph.D.

Academic Editor

PLOS ONE

Additional Editor Comments (optional):

Reviewers' comments:

Reviewer's Responses to Questions

**Comments to the Author**

1. If the authors have adequately addressed your comments raised in a previous round of review and you feel that this manuscript is now acceptable for publication, you may indicate that here to bypass the “Comments to the Author” section, enter your conflict of interest statement in the “Confidential to Editor” section, and submit your "Accept" recommendation.

Reviewer #1: All comments have been addressed

Reviewer #3: All comments have been addressed

2. Is the manuscript technically sound, and do the data support the conclusions?

Reviewer #1: Yes

Reviewer #3: Yes

3. Has the statistical analysis been performed appropriately and rigorously? 

Reviewer #1: Yes

Reviewer #3: Yes

4. Have the authors made all data underlying the findings in their manuscript fully available?

Reviewer #1: Yes

Reviewer #3: Yes

5. Is the manuscript presented in an intelligible fashion and written in standard English?

Reviewer #1: Yes

Reviewer #3: Yes

6. Review Comments to the Author

Reviewer #1: The authors have addressed all issues raised by me in the revised manuscript. The manuscript has been revised accordingly.

Reviewer #3: (No Response)

7. PLOS authors have the option to publish the peer review history of their article (what does this mean? ). If published, this will include your full peer review and any attached files.

**Do you want your identity to be public for this peer review?** For information about this choice, including consent withdrawal, please see our Privacy Policy .

Reviewer #1: **Yes: ** Oche Ochai Agbaji

Reviewer #3: No

---

## [Editor Report · Acceptance letter]

PONE-D-25-05682R1

PLOS ONE

Dear Dr. Siame,

I'm pleased to inform you that your manuscript has been deemed suitable for publication in PLOS ONE. Congratulations! Your manuscript is now being handed over to our production team.

Kind regards,

on behalf of

Dr. Chika Kingsley Onwuamah

Academic Editor

PLOS ONE